# Recent increases in tropical cyclone rapid intensification events in global offshore regions

Yi Li [1,2], Youmin Tang [1,3] ✉, Shuai Wang [4], Ralf Toumi [5], Xiangzhou Song[1,2] & Qiang Wang[1,2]

Rapid intensification (RI) is an essential process in the development of strong tropical cyclones and a major challenge in prediction. RI in offshore regions is more threatening to coastal populations and economies. Although much effort has been devoted to studying basin-wide temporal-spatial fluctuations, variations of global RI events in offshore regions remain uncertain. Here, we show that compared with open oceans, where the annual RI counts do not show significant changes, offshore areas within 400 km of the coastline have experienced a significant increase in RI events, with the count tripling from 1980 to 2020. Furthermore, thermodynamic environments present more favorable conditions for this trend, and climate models show that global ocean warming has enhanced such changes. This work yields an important finding that an increasing threat of RI in coastal regions has occurred in the preceding decades, which may continue under a future warming climate.

Fluctuations in tropical cyclone (TC) activity are a major concern in densely populated coastal regions[1,2]. Although the frequency of TCs has been declining based on observational data and numerical projections[3–5], the threat of intense TCs has continuously increased[6–10]. Hence, accurate predictions are required to improve preparedness and reduce TC-related damage to life and property. Unfortunately, despite improvements in TC track forecasting, errors in intensity forecasting have not substantially decreased in recent decades[11,12]. Rapid intensification (RI) occurs when a TC intensifies dramatically over a short period, thus rendering its forecasting particularly challenging owing to the uncertainty in onset time and duration[13–16]. RI also affects the development of most of the intense TCs[17], as recently demonstrated by Typhoon Hato (2017)[18] and Hurricane Harvey (2017)[19], which caused catastrophic damage. Therefore, detailed examinations of RI are paramount, especially in vulnerable offshore regions.

Previous studies have indicated that the intensification rate of major TCs is increasing in different ocean basins[20,21] and the number of TCs undergoing RI is also rising[22,23]. Shifting of RI locations during recent decades has also been reported in the western North Pacific[21,24] and North Indian Ocean[25]. In recent work, Balaguru et al.[26] showed that the intensification rate increased in coastal areas (defined as within 200 nautical miles from the coast) of the United States, although they did not explicitly analyze the trend of RI events. This warrants additional attention because clear trends have not been defined despite the direct threat posed by RI in offshore regions. Additionally, global surface temperatures have increased at a rate of ~0.18 °C per decade since 1981, which is more than twice that since 1880 (0.08 °C)[27]. In a warming climate, changes in vertical wind shear, mid-level humidity, and ocean temperature are all likely to affect RI and other TC properties[3,28]. At regional scales, the oceanic and atmospheric conditions changes are non-uniform. The faster warming in the western boundary currents[29], for example, might favor TC intensification in these regions[30].

In addition to the ambient environmental vertical wind shear (VWS)[31] and relative humidity (RH)[32] that govern TC intensification, the upper ocean also plays an important role in fueling the overlying atmosphere and TC intensification, as high ocean temperatures favor

[1]College of Oceanography, Hohai University, Nanjing, China. [2]Key Laboratory of Marine Hazards Forecasting, Ministry of Natural Resources, Hohai University, Nanjing, China. [3]University of Northern British Columbia, Prince George, Canada. [4]Department of Geography and Spatial Sciences, University of Delaware, Newark, DE 19716, USA. [5]Department of Physics, Imperial College London, London SW7 2AZ, UK. ✉e-mail: ytang@unbc.ca

TC development by increasing the thermodynamic potential intensity (PI)[33,34]. PI is the stationary maximum intensity after a TC reaches equilibrium, and the changes in its seasonal mean value represent a useful proxy in the analysis of RI long-term variations[35–37]. Instantaneous PI is also an important predictor in statistical prediction systems[13]. Wang et al.[38] developed the concept of potential intensification rate (PIR) and demonstrated that the intensification rate depends on the square of PI and that PIR must be large for RI to occur. Thus, a higher PI in a warming climate favors an increase in both the lifetime maximum intensity (LMI) and global potential TC intensification rate[39]. However, the potential effects of such environmental factors on the variation of RI in offshore regions remain unclear.

In this study, the trends in RI and environmental conditions are analyzed based on the International Best Track Archive for Climate Stewardship (IBTrACS) dataset[40], the fifth generation of the European Centre for Medium-Range Weather Forecasts (ECMWF) reanalysis (ERA5)[41]. We analyze the trends in the counts of RI events across global oceans during 1980–2020, as the TC measurements obtained in this period are reliable because of the wide use of satellite observations and post-season analysis[42,43]. RI represents an increase in the maximum sustained surface wind speed by at least a certain threshold within 24 h. The 45 kt/24 h threshold was used according to recent derivations via objective joint clustering[44]. The results are compared with those obtained using other thresholds, including the more widely used 30 kt/24 h[13]. Similar to Wang and Toumi's study[45], we considered the offshore area to be within 400 km from the nearest landmass larger than 1400 km² (approximately the size of Kauai, Hawaii). Given that the mean translation speed is ~4–5 m/s (350–430 km/day)[46], a TC entering regions within 400 km of a coast would likely make landfall within one

day; hence, these regions represent an urgent concern for operational forecast owing to the relatively short amount of time for predictions and preparedness. Additionally, "near miss" or "indirect-hit" TC tracks can cause significant damage. From a statistical perspective, we show that RI events have exhibited an increasing trend in vulnerable offshore regions on a global scale, and then we subsequently identify the dominant environmental factors. To the best of our knowledge, this study is the first to show such a trend.

## Results

### Trends in RI

The trends in the counts of RI events within 400 km from the coast were first analyzed. The annual number of RI events in these offshore regions during 1980–2020 increased by $3.0 \pm 0.8$ per decade (mean ± standard deviation, Fig. 1a). Over these offshore waters, less than five RI events occurred per year in the 1980s, whereas the annual count increased to ~15 by 2020. Recently, Wang and Toumi[45] reported a significant increase in tropical storm activity in coastal regions. The time fraction of RI periods within the lifespan of a TC has also constantly increased, with a $0.61 \pm 0.14\%$ increase per decade recorded during this 41-year period (Fig. 1b). Therefore, the increase of RI events is likely based on the combination of more TC activity and greater RI probability. Similar trends were observed using the conventional threshold of 30 kt/24 h (Fig. 1c, d), and the annual number of RI events within 400 km to coast increased by $5.1 \pm 1.8$ per decade ($p < 0.01$), with the time fraction also increasing by $0.97 \pm 0.28\%$ per decade. The ratio of RI events over offshore regions relative to the global count has increased, although the trend is not statistically significant ($p = 0.09$, Supplementary Fig. 1).

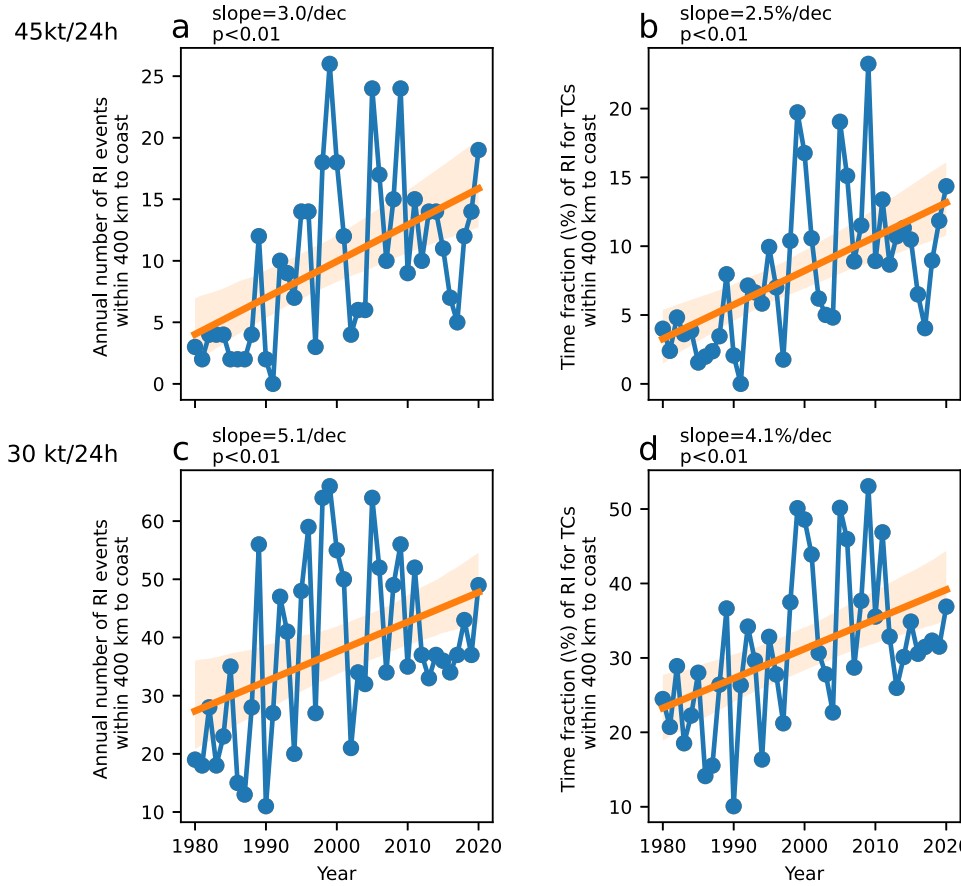

**Fig. 1 | Timeseries of global rapid intensification (RI) activity in offshore regions. a, c** show the annual mean count of RI events within 400 km from the coast. **b, d** show the annual mean time fraction of RI for tropical cyclones within 400 km from the coast. RI is defined as an intensification of at least 45 kt/24 h for **a, b** and 30 kt/24 h for **c, d**. The blue lines and dots show historical data. The orange lines show linear trends, with shading denoting a 95% confidence interval.

RI is generally considered a rare event in the sense that its existence depends on intense TC development. Thus, extreme value analysis is often applied in analyzing trends of such rare events[6,47]. We calculated quantile regressions of the 24-h intensity changes for 90%, 95%, and every 1% quantile between 96% and 99%. Faster increases in 24-h intensity changes were observed over time among the regions within 200–400 km and 400–600 km from the coast, especially for the 97% and higher percentiles (Supplementary Fig. 2). Therefore, the ratio of RI events increased. Given that offshore regions are experiencing more TC activity[45], additional RI events can be expected. An even larger increasing trend was found for ADT-HURSAT (the Advanced Dvorak Technique–HURricane SATellite record) data[48] (Supplementary Fig. 2).

The variability in the annual count of RI events as a function of the distance to land was then calculated (Fig. 2). With an RI threshold of

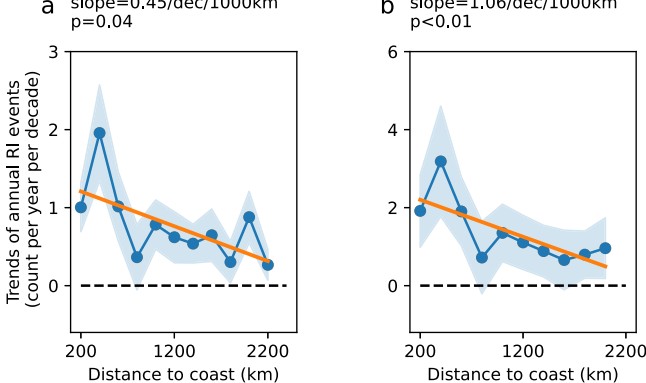

**Fig. 2 | Linear temporal trend of annual rapid intensification (RI) counts for different distance-to-land values.** RI is defined as an increase of at least **a** 45 kt/24 h and **b** 30 kt/24 h. The blue line and dots show linear temporal trends of the annual RI count for each 200-km bin, whereas the blue shading shows the 95% confidence level of the trends. The units are count/decade for the blue line and dots. The orange lines show linear fits of the temporal trends as a function of distance-to-land, and the units are count/decade/1000 km. The slopes and p-values of the orange lines are shown on top of each subplot. The slope is negative, which indicates a landward trend, and the absolute value is shown. The slopes in the following figures are all negative, and their absolute values are shown unless stated otherwise. The x axis is the distance to land from 0–200 km to 2000–2200 km, with a 200-km interval. The y-axis is the temporal trend of the annual count of RI events within each 200-km bin.

45 kt/24 h, the most rapid increase occurred at 200–400 km from the coast, where the annual number of RI increased by 1.9 ± 0.6 per decade (p < 0.01). This temporal trend reduced almost linearly with proximity to land by 0.4 ± 0.2 RI event/decade/1000 km (p = 0.04, Fig. 2a), and the annual number of RI events slightly increased by 0.3 ± 0.2/decade (p = 0.18) when the range extended to 2000–2200 km. As the average annual RI count was 1.6 for each 200-km bin, this distance-dependent variation could reach up to 15%/decade/1000 km or higher. Based on the commonly used RI threshold (at least 30 kt/24 h), a higher and significant temporal trend was detected (1.1 ± 0.3/decade/1000 km, p < 0.01, Fig. 2b). Consistent results were obtained using different thresholds of minimum landmass sizes such as 2000, 5000, and 10,000 km², and different RI threshold including 35 and 40 kt/24 h (Supplementary Fig. 3). The results obtained from the ADT-HURSAT data also showed similar patterns (Supplementary Fig. 4), indicating a valid global-scale landward variation of RI. The climatological distribution shows that on the global scale, RI mainly occurs within 500 km of the coast (Supplementary Fig. 5). The peak shifted from 300 km during 1980–2000 to 400 km during 2000–2020. Nevertheless, the most significant change was within 400 km.

To examine the robustness of the observed landward variation or notable increase in RI events landward, we investigated the data from the pre-satellite period (1951–1979). Although fewer RI events (85.9/year) occurred during this period than during the satellite period (139.6/year), the same landward variation was apparent, with magnitudes of 0.26 to 1.31/decade/1,000 km, depending on the selection of RI threshold (Supplementary Fig. 6).

TC-related damage is dependent on the population and economy exposed. The western North Pacific, Bay of Bengal, Madagascar, Mozambique, Caribbean, and the Gulf of Mexico are usually considered the most vulnerable regions[1]. Significant increases in RI counts were observed in most of these regions (Fig. 3), especially around Madagascar, the South China Sea, and Central America. Specifically, RI events that are higher than 45 kt/24 h in these areas can increase by up to 0.05 (2.5%)/decade for each 2° × 2° latitude–longitude grid, as detected in the Caribbean Sea and along the west coast of Mexico (Fig. 3a and Supplementary Fig. 7). However, RI being a rare event, significant changes vary regionally. The trend in offshore areas discussed above is an overall global effect. Significant increases were observed in the Philippine Sea (130°E–150°E, 10°N–20°N), as reported previously[21,23,24], which is beyond the offshore regions. We thus calculated the landward trend of RI in other regions of the western North Pacific and found that it still shows a significant landward trend

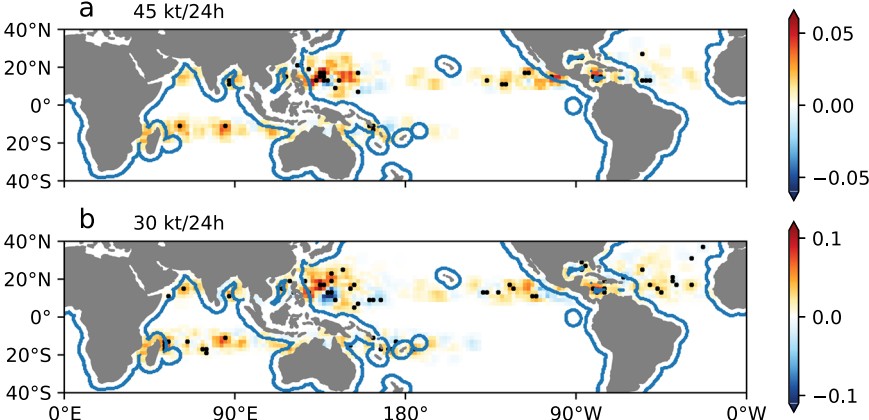

**Fig. 3 | Spatial distribution of the linear trend of annual rapid intensification (RI) counts.** RI is defined as an intensification of at least **a** 45 kt/24 h and **b** 30 kt/24 h. The 41-year linear trend is calculated for each 2° × 2° latitude–longitude grid, and the unit is count/decade. The black dots show areas where 95% confidence for the linear fit is satisfied. The blue lines encompass the regions within 400 km of the coast. Data smoothing using a three-point smoother was performed for better display clarity.

($p = 0.05$, Supplementary Fig. 8). We further show that more RI TC tracks with a 6-h interval occurred within 400 km of the coast, indicating increasing RI processes in offshore regions of this basin (Supplementary Fig. 9). Meanwhile, significant declines were observed in the Central Atlantic and Pacific. When the threshold of 30 kt/24 h was used, similar spatial distributions were observed, except for a patch of decrease located in the Eastern Pacific (Fig. 3b and Supplementary S7). The distance-trend diagrams of individual basins verify the significant increases in RI in coastal regions, except in the Eastern Pacific where increases also occurred over the open ocean, and the North Indian Ocean where the most rapid increase was observed 600–800 km from the coast (Supplementary Fig. 10).

### Influences of climate variability and global warming

As long-term variations in RI are potentially influenced by large-scale circulation and global ocean conditions, the effects of internal climate variability and global ocean warming were investigated. The internal climate variability was assessed using climate indices such as El Niño–Southern Oscillation (ENSO) and Pacific Decadal Oscillation (PDO), while the effects of global warming were estimated using the trend of global mean sea surface temperature (SST) between 60 °S and 75 °N[49]. The impacts of these metrics were eliminated by linearly regressing the time series of the RI onto the climate indices[42]. Marginal differences were found when the Niño3 index was removed, and the amplitude of RI landward variation (Fig. 4a) maintained at $0.4 \pm 0.2$/decade/1000 km. This indicates that the impacts of ENSO on these long-term landward variations are negligible, although it may have a substantial role in inter-annual variability[50]; hence, further exploration is required. Meanwhile, the PDO or global mean SST alone moderately influenced RI. With the removal of their impacts, the landward variation rates decreased slightly to 0.3/decade/000 km for PDO and 0.2/

decade/1000 km for SST (Fig. 4b, c). However, it should be mentioned that with the impacts of global SST removed, the increasing rate of RI events declined to ~0/decade per decade, particularly in regions that were beyond 600 km from the coast. This suggests that global warming is a key driver of increases in RI events; however, the impacts are spatially non-uniform and limited within the open ocean. In contrast, when both the PDO and global SST trend were removed, the landward variation of RI decreased dramatically to $0.05 \pm 0.05$/decade/1000 km ($p = 0.3$, Fig. 4d). These results indicate that the landward trend was insignificant without the effects of PDO and global increases in SST. Therefore, the landward RI variation is likely dominated by both these factors. Other metrics, including the ENSO-Modoki index[51] and North Atlantic oscillation, did not have a substantial influence on RI (Supplementary Fig. 11).

Climate regime shifts have been reported during this 40-year period[52]. An objective detection algorithm[52] was therefore implemented to identify the potential regime shift in RI counts over the offshore regions (Supplementary Fig. 12). Two shifts were found in 1992 and 2002. The annual count of RI events increased in all three regimes, especially from 1992 to 2001. For the RI events over 45 kt/24 h, the increasing rate during 1992–2002 reached 10.7/dec and the average number of RI events increased from 3.4 to 12.4. When the threshold of 30 kt/24 h was used, similar results were also obtained (Supplementary Fig. 12).

### Influences of large-scale environmental factors

High intensification rates and RI formation typically involve large-scale environmental conditions including weak deep VWS, high mid-level RH, and high maximum potential intensity (MPI). Here we show that, RH and VWS became more favorable in offshore areas than in open oceans (Fig. 5), while MPI increased uniformly across most of the globe. The global mean RH value increased by approximately 0.3%/decade/1000 km from the open ocean to offshore region. For VWS, the value decreased in the offshore regions but increased in the open ocean, with a linear trend of $-0.2$ m/s/decade/1000 km ($p < 0.01$). A higher VWS is unfavorable for RI occurrence and thus, the increase in VWS hindered the increase in RI over the open ocean. In addition, more TC activity has been detected in coastal regions[45]. Both landward migration and favorable environments contribute to the increase in RI in offshore regions.

Spatially, both MPI and RH increased in most offshore regions where more RI events occurred (Supplementary Fig. 13), which was also indicated by the correlation between RI and these variables (Supplementary Fig. 14). The rapid increase in MPI over the western South Indian Ocean, South China Sea, western Philippine Sea, and Caribbean Sea coincided with the increase in RI. However, a significant rise in MPI was found in the eastern Philippine Sea (near the Mariana Islands), where a decline in RI events was detected (Fig. 3). Therefore, the correlation between the annual RI count and MPI was negative in this region. In addition, RH showed significant increases and a high correlation with RI in the Arabian Sea, western North Pacific, and Caribbean Sea. As for VWS, ERA5 showed significant changes in the Indian Ocean, South China Sea, and eastern North Pacific, where a high correlation between VWS and RI was observed. The local environmental factors, which were calculated 200–800 km (for VWS and RH) and 200 km (for MPI) from the TC center using 6-h data, were also analyzed. The spatial distribution was strongly affected by the TC activity in each $2° \times 2°$ grid box, resulting in misleading findings. Therefore, the local MPI, VWS, and RH were analyzed for the offshore area; an increasing trend in MPI (5.37 kt/decade, $p < 0.01$) and RH (0.77%/decade, $p = 0.02$) and insignificant trend in VWS (0.14 m/s/decade, $p = 0.12$) were observed. These results suggest that MPI and RH are the dominant environmental conditions in most of the basins, whereas VWS plays an insignificant role in some regions.

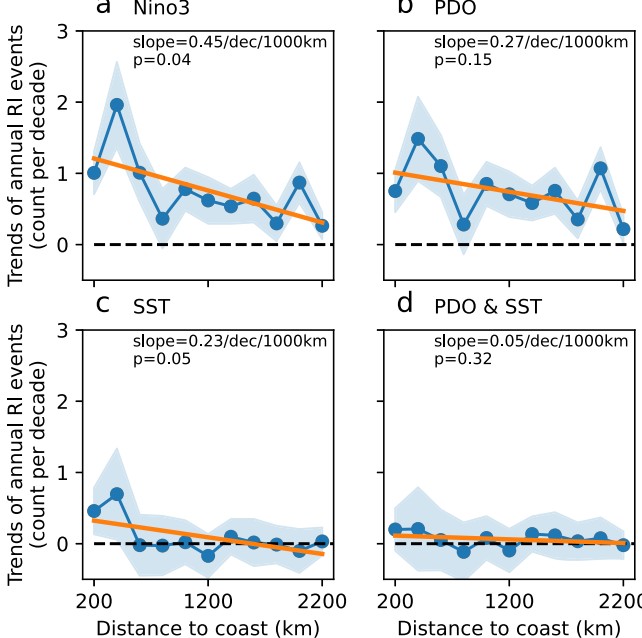

**Fig. 4 | Linear trend of rapid intensification (RI) events with climate indices and/or global sea surface temperature (SST) trend reduced.** Effects of **a** Niño3 index, **b** Pacific Decadal Oscillation (PDO) index, **c** global SST trend, and **d** both PDO and global SST are linearly reduced from the trend of the annual count of RI events. RI is defined as an intensification of at least 45 kt/24 h. The x-axis is the distance to land from 0–200 km to 2000–2200 km, with a 200-km interval. The blue lines and shadings show linear temporal trends and a 95% confidence level of the trends, respectively. The orange lines show linear fits of the temporal trends as a function of distance to land.

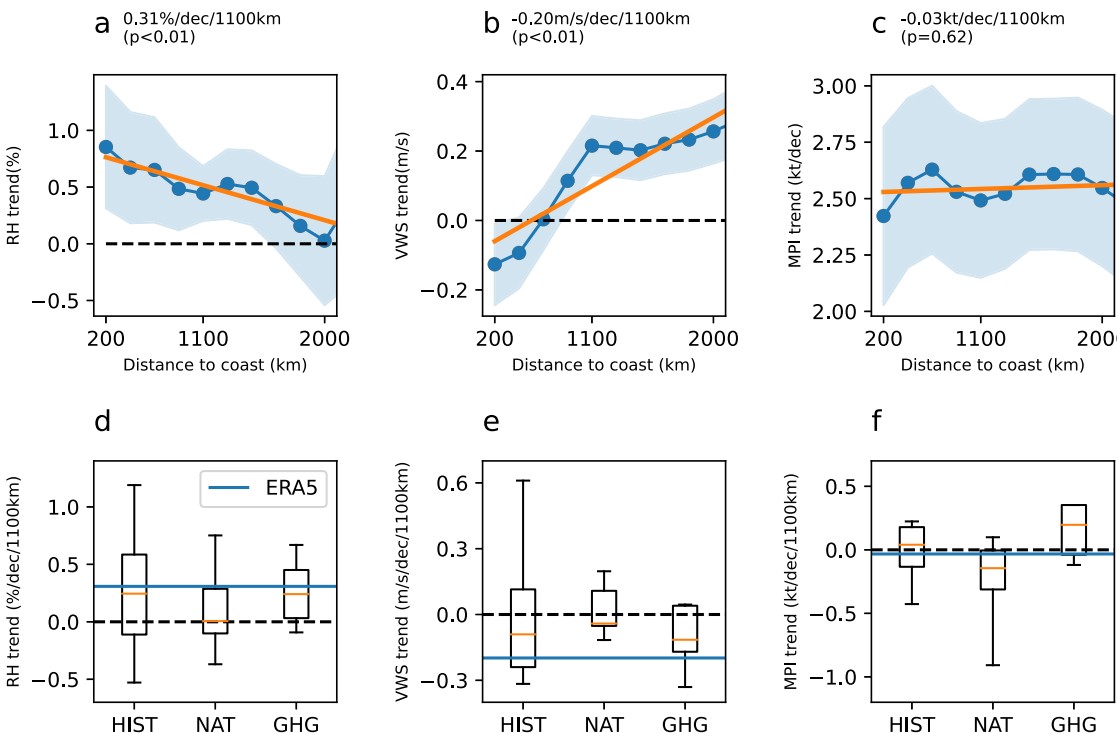

**Fig. 5 | Linear temporal trend of annual mean environment variables for different distance-to-land values. a–c** show the trends of relative humidity (RH), vertical wind shear (VWS), and maximum potential intensity (MPI). The blue line and dots show linear temporal trends of annual mean values for each 200-km bin estimated from ERA5 reanalysis, and the blue shading shows the 95% confidence level of the trends. The orange lines indicate linear fits of the temporal trends as a function of distance-to-land and the slopes and *p* values of the landward trend are shown on top of each subplot. The slopes are reversed, and a positive slope indicates more favorable RH and MPI in the offshore regions, while a negative slope indicates a more favorable VWS in the offshore. The *x* axis shows the distance to land from 0–200 km to 2000–2200 km, with a 200 km interval. The *y* axis is the temporal trend of annual mean values. Only the environmental data during the tropical cyclone season (July to November for the North Hemisphere and December to April for the Southern Hemisphere) were used. **d–f** show the distribution of trends of the MPI, RH, and VWS (i.e., the slopes of the straight lines in **a–c**) simulated in the CMIP6 models. The models were forced with all forcing (HIST), natural variability (NAT), and anthropogenic forcing (greenhouse gas, GHG). The blue horizontal lines show the trend in ERA5 data. The lower and upper ends of the box show the 25th and 75th quartiles, the middle line shows the median, and the horizontal bars below and above the box show the 5th and 95th quartiles, respectively.

The potential intensification rate[38] metric, MPIR, were further analyzed by computing the difference between the current intensity and MPI. MPIR generally follow the distribution of MPI, and the increase can reach up to 10 kt/24 h/decade (Supplementary Fig. 16). However, MPIR significantly decreased in the eastern Philippine Sea, where decreases in RI were also found (Fig. 3). Globally, it increased more rapidly over offshore waters compared with that over open ocean, and the landward slope was 3.7 kt/24 h/decade/1000 km. As the average MPIR was 108 kt/24 h, the increases were ~10%/decade and the landward variation was ~3–4%/decade/1000 km.

Although current climate models cannot directly resolve TC or RI, they provide information on large-scale circulation and therefore can be used to analyze environmental factors. The RH, VWS, and MPI provided by the Coupled Model Intercomparison Project Phase 6 (CMIP6) model were examined to assess the influence of anthropogenic forcing (including aerosols and greenhouse gasses) because both variables showed similar trends as RI. Following Bhatia et al.[37], we used simulations forced with all forcing (HIST), anthropogenic forcing (greenhouse gas, GHG), and natural forcing (NAT) over the period of 1980–2014. For RH, landward trends were detected in most simulations, which are in the same direction as the ERA5 reanalysis, and the slopes were also similar to those in ERA5 (Fig. 5d). In addition, most of the simulated VWS values by the GHG and HIST runs increased from the offshore towards the open ocean, which was consistent with the ERA5 data. In contrast, the NAT runs produced a notably weaker oceanward trend. As for MPI, most of the simulations with all forcings and anthropogenic forcings produced landward MPI trends, while the majority of the NAT simulations showed higher negative trends.

Although a zero-trend was detected in the ERA5 data, the CMIP6 results seemed to indicate that greenhouse gas forcing contributes more to the increase in RI in offshore regions. Discrepancies between the NAT and GHG simulations mainly occurred over the open ocean (Supplementary Fig. 17). For instance, the MPI increased over the eastern Philippian Sea among the NAT runs but decreased significantly in the GHG simulations. MPI is closely linked to SST, which also showed an increasing trend over the eastern Philippines Sea in the NAT runs. However, a higher increase was detected in the offshore regions of the western Pacific in the GHG runs. In addition, the abovementioned analysis was also applied to a longer period from 1950 to 2014 to examine the robustness and steadiness of the aforementioned results over the period from 1980 to 2014. As shown in Supplementary Fig. 18, the results drawn from the longer period are basically consistent with those from the shorter period. MPI, RH, and VWS values derived from the GHG runs showed a more favorable trend than those from the HIST and NAT simulations. Compared with the NAT simulations, GHG shows a more significant landward trend on average in MPI and VWS, with a *p* value of 0.02 and 0.09, respectively, while the different simulations did not produce a significant difference in RH. These results indicate that the trend estimated from the 35-year period is robust and the landward trend of environmental conditions can be potentially attributed more to anthropogenic warming than to internal climate variability.

## Discussion
Forecasting RI is of utmost importance for the prediction of and preparedness for TCs and remains a major challenge in coastal regions.

We analyzed TC observations from 1980 to 2020 and found that RI counts increased significantly more rapidly in offshore regions than in open oceans, in which the combination of the PDO and global ocean warming plays a significant role in this landward variation. These increases can be attributed to the mid-level RH and PI driven by the ocean subsurface temperature in most of the globe, while the VWS initiated significant changes ($p < 0.05$) mainly in the Indian Ocean.

The linear trend analyses in this study are prone to uncertainties, particularly regarding the data and the selected period. Although the best-track data have been used extensively in analyzing TC intensity and variations, the more rapid increase in RI over offshore regions may be an artifact of better observation networks in these regions. To address this, we analyzed ADT-HURSAT data, in which the TC intensity was estimated by processing globally homogenized satellite images using the Advance Dvorak Technique (ADT). Similar landward trends were detected among this dataset (Supplementary Fig. 4). Moreover, an improved least-squares algorithm was applied[53], and it also detected high increases toward the coast (Supplementary Fig. 19). The trend was consistent regardless of the use of the RI threshold (30, 35, 40, or 45 kt/24 h) and the minimum landmass size (1400, 2000, 5000, and 10,000 km$^2$; Supplementary Fig. 3), indicating the robustness of the results. We considered RI events close to either islands or continents in this work because islands are among the most vulnerable regions[1]. However, it would be interesting to separate these two types of landmass because they are distinct in economic development and population density; thus, further explorations are required. Additionally, we only consider the location of RI events, and TCs in the offshore region may not make landfall. Given that a typical TC outer size (radius of gale-force wind) is ~150–200 km, the heat and moisture fluxes continue when the TC center is 400 km away from the coast.

In a recent study, a landward migration of tropical storm activities, such as the LMI and time fraction, was observed due to the enhancement of the westerly steering flow[45]. This landward migration of TC activity indicates a potential increase in RI events. In addition, more favorable environmental conditions, namely higher PI and RH and lower VWS, were found over these regions. CMIP simulations further indicated that anthropogenic forcing has enhanced this landward variation of environmental conditions. However, the spread between CMIP simulations is also considerable and the responses of climate models to difference forcing are also worthy of further investigation. For example, it has been argued that the southwestern Indian Ocean tends to warm more while the western Pacific cools with anthropogenic forcing[54], which probably produces a landward MPI trend in the GHG simulations. However, the underlying mechanism is still ambiguous.

We focused on the RI trends during the past four decades, and the analyses were limited to this period due to the lack of quality of earlier best-track data. Arguably, a 41-year-long period can be insufficient to demonstrate long-term trends or provide evidence of global warming[3]. Regime shifts have been reported during these 41 years[52]. Three regimes were detected for the annual RI count during this period, and the annual mean offshore RI count was significantly higher during the last regime (2002–2020) than during the first (1980–1991). Both global warming slowdown[55] and PDO phase transition[56] potentially affect such regime shifts. For instance, Song et al.[21] reported that the rapid decrease in the PDO index during the early 1990s to early 2000s caused an increase in RI events over the western North Pacific.

Although we only focused on the influence of large-scale environmental factors, both internal and environmental factors contribute to the development of RI[57,58]. By objectively clustering the internal vortex size and intensity, Li et al.[44] developed a threshold of 45 kt/24 h, which was used in this study. However, the observations of inner-core structure and size (usually measured by the radius of maximum wind speed) have not been best-tracked until very recently[59], and analyses of

the long-term trends are not yet feasible. Therefore, conducting further analyses using reliable numerical simulations is necessary. TC activities, including their genesis and tracks, have been examined using climate models and reanalyses[3,5,8]. Although existing models provide evidence of the environmental variations[37], they are still unable to directly resolve RI locations and magnitudes[28]; therefore, models with higher resolution or advanced downscaling techniques are needed[60,61].

Considering the relatively slow progress on improving TC intensity forecasts, the increasing trend of RI in coastal regions poses greater challenges and concerns for operational forecasting. Although the RI trend in climate projections was not explicitly explored, we demonstrated the fundamental role of PI and RH in increased occurrences of RI events and anthropogenic warming drives the landward trend of PI. The projected warming of coastal waters presents an environment with an even more favorable environment and, in turn, more RI events and more challenges in TC prediction. The continuously increasing population and economy in coastal regions also indicate that these areas will potentially have higher exposure and vulnerability to TC threats. Therefore, predictions of and preparation for TCs would be more challenging, and further efforts to improve RI predictions are required.

## Methods
### TC data
TC best-track observations were obtained from the International Best Track Archive for Climate Stewardship (IBTrACS, v4r00)[40], which is supplied with data by US agencies: National Hurricane Center and Joint Typhoon Warning Center. The ADT-HURSAT dataset was developed by processing the globally homogenized HURSAT satellite imagery using the Advanced Dvorak Technique (ADT)[48], and it has been used in several trend analyses[42,46]. Here, we also examined ADT-HURSAT for the period 1980–2017 and compared the results from the two datasets.

We only considered TCs (LMI ≥ 64 kt), over the period 1980–2020. The whole lifecycle of TCs was used. Additionally, to eliminate the influence of topographic effects and extra-tropical transition, we selected only TC tracks over the ocean and within the range of 40°S and 40°N. We only took the records at the standard observational times: 00, 06, 12, and 18 Coordinated Universal Time (UTC). This pre-processing was similar to that performed in the study of RI[13].

In IBTrACS, the default distance to the nearest land, including all continents and islands larger than 1400 km$^2$ (equivalent to the area of Kauai, Hawaii), was provided for each best-track geographical location. We also examined other thresholds of minimum landmass, including 2,000, 5,000, and 10,000 km$^2$, using the coastline data obtained from Global Self-consistent, Hierarchical, High-resolution Geography (GSHHG) database[62].

### Environmental data
The monthly mean atmospheric data, including RH, wind, and atmospheric temperature, were obtained from the fifth generation of ECMWF reanalysis (ERA5)[41]. ERA5 is hosted by the Climate Data Store of the Copernicus Climate Change Service, which regridded the resolution of the ERA5 data to 0.25°. Only the environmental data during TC season (July to November for the North Hemisphere and December to April for the Southern Hemisphere) were used.

### CMIP6 data
Following Bhatia et al.[37], we examined linear trends of the environmental fields from CMIP6 simulations over the period 1950–2014, which was selected because the historical runs spanned until 2014. These fields are defined in an identical way to the observed fields in ERA5. The models include ACCESS-CM2, ACCESS-ESM1,

BCC-CSM2-MR, CanESM5, CESM2, FGOALS-g3, GISS-E2-1-G, HadGEM3-GC31-LL, IPSL-CM6A-LR, MIROC6, MRI-ESM2-0, and NorESM2-LM. All models provide simulations with HIST, NAT, and GHG forcing. The HIST simulations are forced with both GHG and NAT, including volcanoes and solar variability, while the GHG and NAT runs are forced with subsets of the HIST simulations.

### Definitions of RI
The intensification rate was calculated as the change in the maximum surface wind speed ($V_{max}$) in 24 h. RI is commonly defined as an increase in the surface maximum wind speed ($V_{max}$) of at least a threshold within 24 h. The threshold of 30 kt (15.4 m/s), as recommended by Kaplan and DeMaria[13], is widely used. However, other thresholds exist[17], and a physically robust value of 45 kt/24 h was recently derived via objective joint clustering[44]. We used 45 kt/24 h as the major threshold and compared the results using different thresholds.

### Statistical information
The robustness of linear regression usually depends on the choice of period. We, therefore, implemented an improved Ordinary Least-Square (OLS) algorithm[53] and compared the results with those of least-square linear regression. The details of this OLS algorithm are provided in Supplementary Note 1.

### Removal of the global SST trend and climate indices
Following Dai et al.[49] and Kossin et al.[63], the influence of a certain index on the long-term trend could be removed via linear regression from the time series $T$. Let $T(n, i)$ be the number of RI events for year $n$ at the $i$th 200-km bin from the coast and $X(n)$ be the climate index $X$ for year $n$, where $n = 1, 2, ..., 41$ for 1980–2020. The climate indices considered in this study were the global annual mean SST, Niño 3, PDO, ENSO-Modoki, and North Atlantic oscillation. Using a linear regression method, we estimated the trend caused by individual or combination of factors $X$, $T_x(n) = b_x X(n)$, where $b_x$ is the slope in $T(n) = b_x X(n) + \varepsilon$(residual). The residual $\varepsilon$ was then analyzed.

### Environmental variables
The VWS was calculated as the amplitude of wind vector difference between 200- and 850-hPa pressure levels. The mid-level RH was obtained at 600 hPa.

MPI[33,64] was calculated as a function of SST, as follows:

$$\text{MPI} = \sqrt{\frac{\text{SST} - T_o}{T_o} \frac{C_K}{C_D}(k^* - k)}, \tag{1}$$

where $T_o$ is TC outflow temperature determined by the atmospheric vertical profile, $C_D$ the drag coefficient, $C_K$ enthalpy exchange coefficient, $k^*$ the saturation enthalpy of the sea surface, and $k$ the surface enthalpy.

The potential intensification rate (PIR)[38], was calculated from MPI.

$$\text{PIR} = \frac{C_D}{H}\left(E V_{PI}^2 - V^2\right), \tag{2}$$

where $V_{PI}$ is the MPI $V$ the current TC intensity, $H$ the height parameter, and $E$ the dynamical efficiency, obtained using the following equation:

$$E = \left(\frac{f + \frac{2V}{r}}{f + \frac{2V_{PI}}{r}}\right)^n, \tag{3}$$

where $n$ is a sensitivity constant, $r$ is the radius of maximum wind speed, and $f$ is the Coriolis parameter. As suggested by Wang et al.[38], the parameters used were $C_D = 2.4 \times 10^{-3}$, $C_K = 1.2 \times 10^{-3}$, $H = 3$ km, $n = 1$, and $r = 15$ km.

## Data availability

The TC best-track data used in this study were IBTrACS (v4r00) data, retrieved from the NOAA National Centers for Environmental Information (https://www.ncei.noaa.gov/products/international-best-track-archive), and ADT-HURSAT data, obtained from Kossin et al.[48] ERA5 was downloaded from Copernicus Climate Data Store (https://cds.climate.copernicus.eu/#!/home). GSHHG version 2.3.7 was obtained from Paul Wessel[62] (https://www.soest.hawaii.edu/pwessel/gshhg/). Monthly NAO, PDO, and Niño 3.4 indices were downloaded from https://www.cpc.ncep.noaa.gov/products/precip/CWlink/pna/norm.nao.monthly.b5001.current.ascii.table, https://www.ncei.noaa.gov/pub/data/cmb/ersst/v5/index/ersst.v5.pdo.dat, and https://psl.noaa.gov/gcos_wgsp/Timeseries/Data/nino34.long.anom.data, respectively. Global SST data were downloaded from https://www.metoffice.gov.uk/hadobs/hadisst/data/HadISST_sst.nc.gz. CMIP simulations were downloaded from https://aims2.llnl.gov/search. The processed data can also be obtained from https://zenodo.org/record/8115386. Source data are provided with this paper.

## Code availability

The main scripts for data processing and plotting are available at zenodo (https://zenodo.org/record/8115386). Other source codes are available from Yi Li (yli.ouc@gmail.com) upon request.

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

## Acknowledgements
This study was funded by the National Natural Science Foundation of China (Grant nos. 42130409, 42122040, and 42006036), Hohai University (522020512), Singapore Green Finance Centre, Vodafone Foundation and the UK Centre for Greening of Finance and Investment (UKRI-NE/V017756/1).

## Author contributions
Y.L., Y.T., and S.W. conceived the study and analyzed the initial results. R.T., X.S., and Q.W. discussed the results. Y.L. completed the draft, and Y.T., S.W., R.T., X.S., and Q.W. reviewed the manuscript.

## Competing interests
The authors declare no competing interests.
