## [Peer Review File · Nature Communications]

Recent Increases in Tropical Cyclone Rapid Intensification Events in Global Offshore RegionsEditorial Note: This manuscript has been previously reviewed at another journal that is not operating a transparent peer review scheme. This document only contains reviewer comments and rebuttal letters for versions considered at *Nature Communications*.

REVIEWER COMMENTS

Reviewer #2 (Remarks to the Author):

Although the authors give some analyses on the different influences of global warming and natural variability by using CMIP6 products, they fail to represent a reasonable explanation. I have to give a major revision again.

The PDO is the dominant mode on multidecadal timescales over the Pacific. The current CMIP6 timespan (1980-2014) is too short to analyze the PDO-related signal, since it includes only one warm PDO phase and one cold PDO phase. Note that there have been some CMIP6 outputs in a longer period (e.g., 1950–2014; Agel and Barlow, 2020). As the authors noted, it is quite hard to identify RI events in the CMIP6 due to a coarse resolution. But the analysis on environmental conditions during a longer period can provide some useful information.

Besides, after the authors adding the offshore lines, it is clear that greater and more significant MPI increases are mainly observed over the open sea, especially outside the defined offshore regions (Figure S14). I wonder why Figure 5 gives a greatest MPI increase for the distance to land within 0-200 km.

I also notice that most of the NAT runs show negative MPI slopes in Figure 5C, which is opposite compared to the GHG runs. How do the authors explain this discrepancy? Would the authors provide the spatial distribution of MPI changes for the ensemble mean of NAT/GHG runs?

Finally, the CMIP6 outputs do not capture the changes in RH, compared with the ERA5.

Reference

Agel, L., and M. Barlow, 2020: How Well Do CMIP6 Historical Runs Match Observed Northeast U.S. Precipitation and Extreme Precipitation–Related Circulation?. *J. Climate*, 33, 9835–9848, <https://doi.org/10.1175/JCLI-D-19-1025.1>.

Reviewer #2 (Remarks to the Author):

Although the authors give some analyses on the different influences of global warming and natural variability by using CMIP6 products, they fail to represent a reasonable explanation. I have to give a major revision again.

Dear Reviewer,

We appreciate your valuable comments very much. We understand your concerns and apologize for any confusion. As suggested, we have provided an additional analysis using CMIP6 simulations over a longer period (1950–2014) to address the discrepancy and uncertainty. We have also added an in-depth discussion to alleviate your other concerns to the best of our ability. Our point-by-point responses are provided below (your comments are in black, and our reply is in blue). We hope that you will now find our paper acceptable for publication.

The PDO is the dominant mode on multidecadal timescales over the Pacific. The current CMIP6 timespan (1980-2014) is too short to analyze the PDO-related signal, since it includes only one warm PDO phase and one cold PDO phase. Note that there have been some CMIP6 outputs in a longer period (e.g., 1950–2014; Agel and Barlow, 2020). As the authors noted, it is quite hard to identify RI events in the CMIP6 due to a coarse resolution. But the analysis on environmental conditions during a longer period can provide some useful information.

Thank you very much for this comment. As you suggested, we have added an analysis based on CMIP6 simulations over the 65-year period from 1950 to 2014. Overall, the results from this long period are consistent with those obtained for the period 1980 to 2014, as shown in Fig R1 and Fig. S18. Namely, the GHG simulations produced a more favorable environment (i.e., positive landward trend) than the HIST and NAT runs, indicating the robustness of the previous results.

Figure R1 (Figure S18). Distribution of trend of the (a) MPI, (b) RH, and (c) VWS simulated in CMIP6 models over the period of 1950–2014. The models were forced with all forcing (HIST), natural variability (NAT), and anthropogenic forcing (greenhouse gas, GHG). A positive trend in the MPI and RH or a negative trend in the VWS indicates a more favorable environment in the offshore regions. The lower and upper ends of the box show the 25th and 75th quartiles, the middle line shows the median, and the horizontal bars below and above the box show the 5th and 95th quartiles, respectively.

We have added more results and discussion in the revised manuscript:

Lines 258-265: In addition, the abovementioned analysis was also applied to a longer period from 1950 to 2014 to examine the robustness and steadiness of the aforementioned results over the period from 1980 to 2014. As shown in Fig. S18, the results drawn from the longer period are basically consistent with those from the shorter period. MPI, RH, and VWS values derived from the GHG runs showed a more favorable trend than those from the HIST and NAT simulations. Compared with the NAT simulations, GHG shows a more significant landward trend on average in MPI and VWS, with a p -value of 0.02 and 0.09, respectively, while the different simulations did not produce a significant difference in RH. These results indicate that the trend estimated from the 35-year period is robust and the landward trend of environmental conditions can be potentially attributed more to anthropogenic warming than to internal climate variability.

Besides, after the authors adding the offshore lines, it is clear that greater and more significant MPI increases are mainly observed over the open sea, especially outside the defined offshore regions (Figure S14). I wonder why Figure 5 gives a greatest MPI increase for the distance to land within 0-200 km.

Thanks for this comment. This discrepancy is because in the original Figure 5, the MPI trends were calculated based on the counts of TC activity in each 2° by 2° pixel, while in

Figure S14, the spatial distribution was computed using only the average MPI within each pixel. As you pointed out, the method using the counts of TC activity might not be appropriate. Therefore, we recalculated the trend based on the spatial distribution only, and obtained consistent results (a flat trend in the MPI as shown in Figure R2c and Figure 5c).

Figure R2 (Figure 5). Linear temporal trend of annual mean environment variables for different distance-to-land values. a, b and c show the trends of relative humidity (RH), vertical wind shear (VWS), and maximum potential intensity (MPI). The blue line and dots show linear temporal trends of annual mean values for each 200-km bin estimated from ERA5 reanalysis data, the blue shading shows the 95% confidence level of the trends, and the orange lines indicate linear fits of the temporal trends as a function of distance-to-land and the slopes. *p*-values of the landward trend are shown on top of each subplot. The slopes are reversed, and a positive slope indicates more favorable RH and MPI in the offshore regions, while a negative slope indicates a more favorable VWS in the offshore. The x-axis shows the distance to land from 0–200 km to 2,000–2,200 km, with a 200-km interval. The y-axis is the temporal trend of annual mean values. Only the environmental data during the tropical cyclone season (July to November for the North Hemisphere and December to April for the Southern Hemisphere) were used. d, e and f show the distribution of trends of the MPI, RH, and VWS (i.e., the slopes of the straight lines in a, b and c) simulated in the CMIP6 models. The models were forced with all forcing (HIST), natural variability (NAT), and anthropogenic forcing (greenhouse gas, GHG). The blue horizontal lines show the trend in ERA5 data. The lower and upper ends of the box show the 25th and 75th quartiles, the middle line shows the median, and the horizontal bars below and above the box show the 5th and 95th quartiles, respectively.

We have clarified this information in the revised manuscript.

Lines 194-195: Here we show that, RH and VWS became more favorable in offshore areas than in open oceans (Fig. 5), while MPI increased uniformly across most of the globe.

Lines 199-200: In addition, more TC activity has been detected in coastal regions⁴⁷. Both landward migration and favorable environments contribute to the increase in RI in offshore regions.

I also notice that most of the NAT runs show negative MPI slopes in Figure 5C, which is opposite compared to the GHG runs. How do the authors explain this discrepancy? Would the authors provide the spatial distribution of MPI changes for the ensemble mean of NAT/GHG runs?

We added spatial variations, as suggested. The results show that for GHG simulation, the MPI were observed to have an increase in the offshore regions of South China Sea and southwestern Indian Ocean but a decrease in the tropical Atlantic and western North Pacific (Figure R3 and S17). The spatial distribution of MPI is mainly dominated by the SST distribution since the former is directly linked to the latter in definition. Fig R3 compares the linear trend between the MPI and SST, showing their high consistence. Thus, the non-uniform ocean warming causes such a MPI distribution, which has also been found in literature. For example, Dong et al (2004) found that anthropogenic forcing produces an IOD-like warming pattern in the Indian Ocean and the southwestern Indian Ocean warms but a cooling in the western North Pacific. The reason responsible for such a non-uniform ocean warming under the background of global warming is complex topic and beyond the scope of this study.

Fig. R3 (Fig. S17). Spatial distribution of the linear trend of MPI and SST estimated using the mean of (A) and (B) NAT and (C) and (D) GHG simulations in CMIP6. The black dots show areas where the 95% confidence interval of the linear fit was satisfied. The blue lines encompass the regions within 400 km from the coast. The trends were calculated for each 2 by 2 longitude-latitude grid, and data smoothing using a three-point smoother was performed for better display clarity.

We have added the following results and discussion:

Lines 253-257: Discrepancies between the NAT and GHG simulations mainly occurred over the open ocean (Fig. S17). For instance, the MPI increased over the eastern Philippian Sea among the NAT runs but decreased significantly in the GHG simulations. MPI is closely linked to SST, which also showed an increasing trend over the eastern Philippines Sea in the NAT runs. However, a higher increase was detected in the offshore regions of the western Pacific in the GHG runs.

Lines 291-295: However, the spread between CMIP simulations is also considerable and the responses of climate models to different forcings are also worthy of further investigation. For example, it has been argued that the southwestern Indian Ocean tends to warm more while the western Pacific cools with anthropogenic forcing⁵⁵, which probably produces a landward MPI trend in the GHG simulations. However, the underlying mechanism is still ambiguous.

Finally, the CMIP6 outputs do not capture the changes in RH, compared with the ERA5.

It is true that CMIP6 models do not capture the change in RH well. This is probably because of the difficulty in simulating RH in climate models owing to its high sensitivity to meteorological factors and large uncertainties in parameterizing relevant atmospheric processes, such as atmospheric convection and condensation. Thus, it is advised to interpret the RH results with caution. We have included additional CMIP6 simulations to estimate the RH trend, and the updated CMIP6 data are more consistent with the reanalysis.

We have revised the following sections:

Lines 245-247: For RH, landward trends were detected in most simulations, which are in the same direction as the ERA5 reanalysis, and the slopes were also similar to those in ERA5 (Fig. 5d).

Reference:

Bhatia, K., Baker, A., Yang, W., Vecchi, G., Knutson, T., Murakami, H., et al. (2022). A potential explanation for the global increase in tropical cyclone rapid intensification. *Nature Communications*, 13(1), 6626. <https://doi.org/10.1038/s41467-022-34321-6>

Dong, L., Zhou, T., & Wu, B. (2014). Indian Ocean warming during 1958–2004 simulated by a climate system model and its mechanism. *Climate Dynamics*, 42(1–2), 203–217.

<https://doi.org/10.1007/s00382-013-1722-z>

Wang, S., & Toumi, R. (2021). Recent migration of tropical cyclones toward coasts. *Science*, 371(6528), 514–517. <https://doi.org/10.1126/science.abb9038>

REVIEWERS' COMMENTS

Reviewer #2 (Remarks to the Author):

I am sorry for the delay of my comments. I appreciate the authors' effort on revising this manuscript. All of my previous concerns are reasonably replied. I have just one more minor suggestion. In Figure 3, the green and blue lines are not easy to distinguish. I would suggest the authors use different colors or shadings.

Reviewer #2 (Remarks to the Author):

I am sorry for the delay of my comments. I appreciate the authors' effort on revising this manuscript. All of my previous concerns are reasonably replied. I have just one more minor suggestion. In Figure 3, the green and blue lines are not easy to distinguish. I would suggest the authors use different colors or shadings.

Dear Reviewer,

We appreciate your time and effort very much. Your comments and suggestions have greatly helped improve the manuscript. As suggested, we have changed the color of coastline in Figure 3 and use grey patches to represent the land. The figure is much clearer now.

Thank you again.